# Preparation and Characterization of Wood Scrimber Based on Eucalyptus Veneers Complexed with Ferrous Ions

**DOI:** 10.3390/polym14194217

**Published:** 2022-10-08

**Authors:** Shiqin Liu, Qiuqin Lin, Yanglun Yu, Wenji Yu

**Affiliations:** Research Institute of Wood Industry, Chinese Academy of Forestry, Beijing 100091, China

**Keywords:** eucalyptus, wood scrimber, ferrous ions, complexes, surface properties

## Abstract

Wood-based products manufactured from fast-growing wood species such as eucalyptus have gained increasing attraction with the demand of using wood in architecture, furniture, and decoration. In this paper, a new type of wood scrimber based on eucalyptus veneers complexed with ferrous ions was prepared and its properties were characterized. The results showed that the presence of complexes did not affect the mechanical properties of eucalyptus wood scrimber, but made its surface more hydrophobic (contact angle increased by 38.48% and dimensional stability improved (thickness swelling rate decreased by 32.26%). Most importantly, the color of eucalyptus wood scrimber changed significantly, from the original brown to dark blue, and its anti-photoaging property also greatly improved. These advantages would make this type of wood scrimber based on the eucalyptus veneer complexes with ferrous ions more widely applicable in decorations and buildings.

## 1. Introduction

As one of the three fastest-growing plantation species, eucalyptus is planted in more than 90 countries with over 20 million hectares of plantations [1] and is also abundant in resources and distributed in many provinces in South China, such as Guangdong and Guangxi province, with an area of more than 20,000 cubic kilometers [2,3]; thus, eucalyptus plantations are usually used to produce wood chips for paper, pulp industry and furniture manufacturing [4]. However, the poor physical and mechanical properties of the fast-growing forest due to the low degree of lignification [5,6], coupled with the small diameter class, high growth stress, and natural small defects of eucalyptus, seriously affect its mechanical and visual characteristics, and limit its large-scale and high value-added applications [7]. Many eucalyptus-based engineered products with high quality have been proposed in recent years such as laminated veneer lumber (LVL), cross-laminated timber (CLT), or oriented strand lumber (OSL). However, these wood products based on eucalyptus have problems, including end crack, veneer fracture, and bonding difficulties due to poor adhesive permeability and high internal stress [8,9].

Therefore, the mechanical and physical properties of structural eucalyptus wood scrimber [10] obtained can be improved by using a novel combination technology for reference. Then, the problems of small diameter class, too many defects, bonding difficulties in the manufacturing process and large-scale application of eucalyptus can be solved and the high value-added utilization of eucalyptus is also realized [11,12]. Since the properties of eucalyptus wood scrimber obtained using the novel combination technology are controllable and its specifications are adjustable [10], it is widely used as indoor decoration and outdoor structural materials.

Eucalyptus extracts are rich in plant polyphenols [13,14], which have attracted considerable interests due to their important biological activities as well as intriguing chemical and physical properties. Coordination complexing with metal is an important property of polyphenols [15,16]. The complexing reaction between polyphenols and metal can be regulated by the kinds of metal ions [17,18], the concentration of metal salts [19,20] and the pH of solution [21]. Most importantly, polyphenols can complex with various metal ions to form different complexes of different colors which indicates the various colored eucalyptus wood scrimbers can be obtained by treating with different metal ions. It may not only maintain the excellent performance of traditional eucalyptus wood scrimber, but also improve some practical properties of it to a certain extent and obtain more color-optional and decorative eucalyptus wood scrimber.

Herein, eucalyptus in this paper was used as raw material and ferrous ion was used as a metal complexing agent; a new kind of eucalyptus wood scrimber was prepared by vacuum impregnating the ferrous sulfate solution of the recombined eucalyptus veneers to in situ fix the polyphenols in the wood cell itself in the original steps of the preparation of traditional wood scrimber. Then, the surface properties, color index, anti-photoaging property, water absorption, dimensional stability and mechanical properties of the prepared Fe (II)–complex eucalyptus wood scrimber were tested and analyzed to explore the feasibility and diversity of it as decoration, furniture, and structure materials, thus to promote its utilization, application and promotion.

## 2. Materials and Methods

### 2.1. Materials

Eucalyptus logs purchased from the Guangxi province of Northeast China were used as raw materials (60% moisture content, 15~20 cm diameter, and 0.65 g/cm^3^ density). The eucalyptus veneers with a thickness of 6 mm were peeled from the round wood, and a series of linear cracks which constitute oriented wood fiber mats were made along their longitudinal direction by a self-made fluffing machine. The oriented wood fiber mats were then dried in the kiln until the moisture content decreased to about 8~10%.

Phenol formaldehyde (PF) resin was purchased from Guangdong Dynea Chemical Industry Co (45% solid content, 45 cps viscosity, and pH 9–10). Ferrous sulfate heptahydrate (technical analysis, 96%) was purchased from Guangxi province of China.

### 2.2. Material Preparation of Fe (II)-Complex Eucalyptus Scrimber

Figure 1a–g show the preparation process of the Fe (II)–complex eucalyptus scrimber. This involved cutting the prepared oriented wood fiber mats to 450 mm length and vacuum impregnating them with 80,000 mL of FeSO_4_·7H_2_O solution at different concentrations (1.5, 2.5, and 3.5 wt%) in a vacuum pressure impregnation device (SBK-450B-000920) for about 24 h, then drying to 10% MC in natural atmosphere. Those eucalyptus fiber mats were immersed in the FP resin to achieve 18% resin content and dry again to 8% MC. The eucalyptus fiber mats were weighted according to the setting density of about 1.15 g/cm^3^ and then paved along the fiber’s longitudinal direction in the mold. After that, they were hot-compressed at 170 °C and 5 MPa, and the hot-pressure process lasted for 15 min. The obtained Fe (II)–complex eucalyptus scrimbers with dimensions of 450 mm × 160 mm × 15 mm were conditioned in a chamber at 65% ± 5% relative humidity (RH) at (20 ± 2) °C for 2 weeks before the following testing. The control sample is denoted as eucalyptus wood scrimber (EWS) and the treated samples are denoted as 1.5%Fe@EWS, 2.5%Fe@EWS and 3.5%Fe@EWS, respectively.

### 2.3. Characterization

The microstructure morphological features of a cross-section of the wood scrimber with a different concentration of FeSO_4_·7H_2_O were observed using a scanning electron microscope (SEM) (KEYENCE MSD-VHX1000), coupled with an energy dispersed X-ray (EDX) analyzer (Horivba 7021-H, Kyoto, Japan). The SEM and EDX images were acquired with an acceleration voltage of 10 kV. The surface roughness was measured using scanning probe microscopy (SPM, TIME3230, Beijing Time Peak Technology Company, Beijing, China) and an EDF 3D microscope (VHX-6000, Keyence, Osaka, Japan). The magnification of the objective lens used in the method was 100× and the scan size was 10 mm × 10 mm.

### 2.4. Surface Color Testing

Color changes were measured using a chroma meter (CR-400, K & M Co., Tokyo, Japan) and D65 light source. The color was assessed with the aid of the color space CIEL × *a***b** with coordinates representing the lightness of the given color (from *L** = 0 (black) to *L** = 100 (white)), its position between red and green (*a**, negative values indicate green, and positive values indicate red), and its position between yellow and blue (*b**, negative values indicate blue, and positive values indicate yellow). The color changes were expressed through partial differences
(1)$$C^{*}=\sqrt{{\left(a{\;}^{*}\right)}^{2}+{\left(b{\;}^{*}\right)}^{2}}$$
(2)$$\Delta L^{*}=L_{2}-L_{1}$$
(3)$$\Delta a^{*}=a_{2}-a_{1}\;$$
(4)$$\Delta b^{*}=b_{2}-b_{1}$$
with Equations (2)–(4) indicating the lightness values, red-green values and blue-yellow values, respectively, before and after ferrous sulfate treatment, and the color saturation difference Δ*C**, color difference Δ*H**, and overall color difference Δ*E** were calculated according to Equations (5)–(7), respectively, [22,23]. Each sample was assessed at five points, and the average value was used in the data analysis.
(5)$$\Delta C^{*}=C_{2}-C_{1}$$
(6)$$\Delta E^{*}=\sqrt{{\left(\Delta L^{*}\right)}^{2}+{\left(\Delta a^{*}\right)}^{2}+{\left(\Delta b^{*}\right)}^{2}}$$
(7)$$\Delta H^{*}=\sqrt{{\left(\Delta E^{*}\right)}^{2}-{\left(\Delta L^{*}\right)}^{2}-{\left(\Delta C^{*}\right)}^{2}}$$

The anti-photoaging property testing was evaluated by Xenon Test Chamber (XE-2-HS) under irradiance of 1.10 W/m^2^ 420 nm, temperature at 63 °C, relative humidity of 50%, according to GB/T 17657-2013. *L**, *a**, *b**, and Δ*E** were measured after 96 h testing.

### 2.5. Water Absorption and Dimensional Stability Tests

The water absorption rate (WAR, defined as WAR in the equation), thickness swelling rate (TSR, defined as TSR in the equation) and width swelling rate (WSR, defined as WSR in the equation) of the wood scrimbers were tested according to ASTM D-1037-12 “test methods for evaluating properties of wood-base fiber and particle panel materials” and were calculated using Equations (8)–(10), respectively. In detail, 6 replicate samples with dimensions of 50 mm × 50 mm × 16 mm were completely immersed in boiling water for 4 h, and then dried in an oven at 60 ± 3 °C for 20 h. After that, those samples were immersed into boiling water for another 4 h again and then taken out with the removal of excessive water. The water absorption was determined by the weight percent gain before and after the testing. The thickness and width swelling rate were calculated based on the mid-span thickness and width changes.
(8)$$W_{AR}=\frac{m_{2}-m_{1}}{m_{1}}\times 100\%$$
where *m*_1_ (g) and *m*_2_ (g) are the weight of the samples before and after the testing, respectively.
(9)$$T_{SR}=\frac{T_{2}-T_{1}}{T_{1}}\times 100\%$$
where *T*_1_ (mm) and *T*_2_ (mm) are the thickness of the samples before and after the testing, respectively.
(10)$$W_{SR}=\frac{W_{2}-W_{1}}{W_{1}}\times 100\%$$
where *W*_1_ (mm) and *W*_2_ (mm) are the width of the samples before and after the testing, respectively.

### 2.6. Mechanical Tests

The modules of elasticity (MOE) and the modules of rupture (MOR) were measured in accordance with the GB/T 17657—2013 “Test methods of evaluating the properties of wood-based panels and surface decorated wood-based panels”. There were six samples for the bending test and six samples for the horizontal shear strength test with dimensions of 280 mm (longitude) × 20 mm (width) × 12 mm (thickness), respectively.

## 3. Results and Discussion

### 3.1. Morphologies of Cross-Section of Wood Scrimber

Figure 2 showed the EDX analysis of Fe@EWS and EWS. As seen from the EDX images, the Fe element was evenly distributed in wood cells and the higher the ferrous sulfate concentration used, the more iron atoms immobilized on the cross-section of the wood scrimbers (a3–d3). In the case of ferrous sulfate with 3.5% concentration, the atomic ratio and mass fraction ratio of iron immobilized on the surfaces reached 2.15% and 6.54%, respectively. This indicated that the eucalyptus wood fiber mats can absorb Fe (II) ions from ferrous sulfate aqueous solutions.

The cross-section morphologies of wood scrimber were characterized using SEM. Figure 3 shows the cross-section SEM images of the wood scrimber with different Fe (II) concentration. Particle clusters sometimes could be found in some cell cavities (b) and wood rays (c,d) compared with the control samples (a). This suggested that some of the iron complexes attached to the wood scrimbers existed as particles in addition to being tightly bound to the cell wall. Moreover, with the increase in iron concentration, the number of particles rose, and the size of particle size became smaller.

### 3.2. Surface Color and Anti-Photoaging Property

As revealed in previous reports, eucalyptus extractives mainly included polyphenols which have a significant influence on the color of raw materials [24,25,26]. Plant polyphenols can mostly be divided into hydrolyzed tannins and condensed tannins [15]. Hydrolyzed tannins contained chromogenic groups. Additionally, catechins, the parent of condensed tannins, can be used as natural dyes after oxidation and condensation. The polyphenols react with iron ions to form dark complexes which are insoluble in water [27,28,29]. For example, hydrolyzed tannins include gallotannins and ellagitannins which can be hydrolyzed to glucose and gallic acid or ellagic acid, respectively [30]. Combining with ferrous ions, hydrolysable tannins form blue-black-colored Fe (III)–tannin complexes since the reversible charge transfers across the Fe (III)-O bond in the Fe (III)–tannate complex [31]. Condensed tannins (proanthocyanidins) were oligomers or polymers of flavan-3-ol (catechin) monomers which, combined with Fe (III), form green-black-colored complexes [32,33].

In order to reveal the effect of Fe (II) ions on the color change of eucalyptus wood scrimber, the changes in CIELab color system for EWS, 1.5%Fe@EWS, 2.5%Fe@EWS, and 3.5%Fe@EWS were explored. As shown in Figure 4a, when the concentration of ferrous sulfate reached 3.5%, the values of chromaticity coordinate *a** for the wood scrimber dropped to −0.77, a 110.8% decrease, as compared to the values of their control samples. At the same time, when the values of chromaticity coordinate *b** and *L** for the wood scrimber reached their minimum, the corresponding concentrations of ferrous sulfate were 1.5% and 3.5%, 74.9% and 33.55% decreases, respectively. All of these indicate that FeSO_4_ pretreatment had a great influence on the color of eucalyptus wood scrimber. Moreover, the numerical increases in chromaticity coordinate *a** and *b** with increasing Fe (II) suggested that the color of the wood scrimber gradually became greener, bluer and darker.

As can be seen in Figure 4b, when the concentration of Fe (II) increased from 1.5% to 3.5%, the color saturation difference Δ*C**, the color difference Δ*H** and the overall color difference Δ*E** of Fe (II)–complex wood scrimbers increased from −16.42, 3.34 and 22.95 to −15.24, 5.27 and 24.81, respectively. The wood scrimber with a different concentration of Fe (II) presented a similar increasing trend in Δ*C**, Δ*H**, and Δ*E**, indicating that the color of wood scrimber gradually changed from a primary color to a dark color, the brightness of the wood scrimber gradually decreased, and the color gradually deepened.

Figure 5 shows the chromaticity changes of the EWS and 1.5%Fe@EWS samples after 96 h of irradiation in a xenon light attenuator. The color of eucalyptus wood the scrimber began to change after artificially accelerated aging with a xenon lamp, resulting in a color difference. As can be seen from Figure 5, after 96 h of artificially accelerated aging with a xenon lamp, the chromaticity index *L** (a) of the 1.5%Fe@EWS decreased by 1.64 and 2.33, respectively. Both chromaticity index *a** (b) and *b** (c) of EWS and 1.5%Fe@EWS showed an upward trend, increasing by 0.862 and 4.01, 0.45 and 2, respectively. The decrease in *L** values and the increase in *a** and *b** values indicated that the brightness of the wood decreased and the color tended to be darker and yellower. The overall color difference Δ*E** of EWS was 5.8 while that of 1.5%Fe@EWS was just 2.86, a decrease of 50.70% (d). Both of them were judged as the third-grade standard of blue wool by GB/T 17657-2013. It can be seen that the complex formed by ferrous ions and polyphenols in eucalyptus had the effect of inhibiting photochromism, reducing the overall color difference Δ*E** of the wood scrimber and enhancing the anti-photoaging property.

### 3.3. Surface Roughness and Wettability

As can be seen from Figure 6, the surface of the eucalyptus wood scrimbers were mainly composed of a longitudinal arrangement of fibers and cured adhesive. The height of the surface topography of the samples can be clearly seen from the three-dimensional topography of the ultra-depth-of-field microscope, and the surface topography and roughness of both EWS and 2.5%Fe@EWS had the same trend. Compared with the EWS, the Ra and Rz values of 2.5%Fe@EWS increased by 175.65% and 92.09%, respectively. At the same time, the surface water contact angle increased from 69.47 to 96.20. These results showed that the surface contact angle of eucalyptus wood scrimber increased and the surface wettability of them decreased after complexing with Fe (II).

### 3.4. Water Absorption and Dimensional Stability

WSR, TSR, and WAR are three important indexes for evaluating the dimensional stability and water absorption of wood products [10]. Here, the samples tended to recover better along the thickness direction than that of width because the thickness was the compression direction [34]. Therefore, the WSR of all samples was much lower than TSR because the thickness was the compression direction, as shown in Figure 7. For all conditions, the TSR and WSR of EWS@Fe decreased compared with that of EWS. There was a 2.05%, 32.26%, and 25.88% reduction in the TSR of 1.5%Fe@EWS, 2.5%Fe@EWS, and 3.5%Fe@EWS in the 28 h cycle test, respectively. The WAR showed an increase for 3.30% in the case of 2.5%Fe@EWS while downtrends were shown in 1.5%Fe@EWS and 3.5%Fe@EWS, 5.86% and 7.75%, respectively. These results indicated that both the dimensional stability and water absorption of wood scrimber had a relative improvement after coordination complexing with ferrous ions.

### 3.5. Mechanical Properties

The modules of rupture (MOR) and modules of elasticity (MOE) of round wood and wood scrimber with different concentrations of Fe (II) are shown in Figure 8. In Figure 8 a, the modules of rupture of the wood scrimber with or without Fe (II) ions basically increased compared with that of the eucalyptus round wood. The MOR of Fe@EWS decreased compared with the EWS samples, and the lowest reduction rate of 2.5% Fe@EWS was 8.16%, which was also 35.45% higher than that of round wood. As can be seen from Figure 8b, the MOE increased slightly (5.16%) when the iron concentration was 2.5% with a maximum of 23,432 MPa compared with EWS at the same time. This was because MOE was a basic property of materials, which was related to the types of materials. Therefore, there were barely changes along with the iron concentration. All these wood scrimbers presented a mechanical property superior to round wood.

## 4. Conclusions

The eucalyptus Fe (II)–complex wood scrimber in this study was successfully prepared and high surface properties and dimensional stability were obtained. The impregnated ferrous sulfate was inserted into the fiber cell walls, forming Fe (II)-polyphenol complexes, which improved the water contact angle by 38.48% and reduced water absorption and thickness swelling by 7.75% and 32.26%, respectively. Additionally, after ferrous sulfate treatment, the surface color of eucalyptus wood scrimber also changed significantly, showing a trend of being darker, greener and bluer. At the same time, the anti-photoaging property of it also improved greatly via the 96 h of xenon lamp aging test. However, these properties were affected by the concentration of Fe (II). With 2.5% Fe contents, the wood scrimber had a good MOE and MOR. Those surface properties, dimensional stabilities and mechanical properties with high added value suggested that the eucalyptus Fe (II) complex wood scrimber can be used for structural materials and decorated materials such as natural-colored floor.

## Figures and Tables

**Figure 1 polymers-14-04217-f001:**
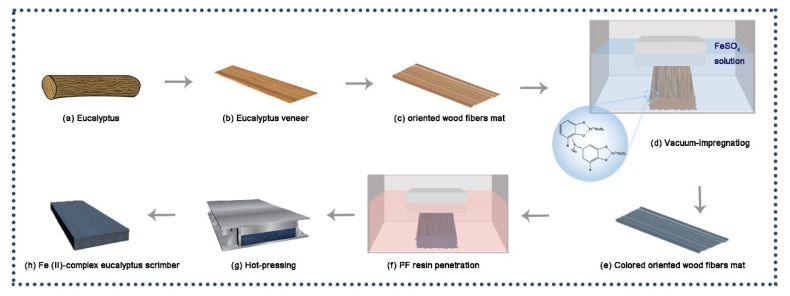
Preparation process of Fe (II)–complex wood scrimber (**a**–**h**).

**Figure 2 polymers-14-04217-f002:**
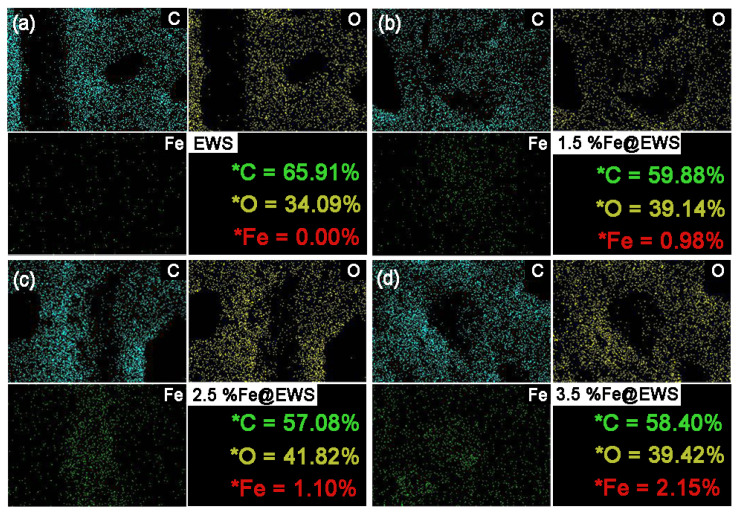
EDX images of Control samples (EWS) (**a**), 1.5%Fe@EWS (**b**), 2.5%Fe@EWS (**c**), and 3.5%Fe@EWS (**d**). The number in the figure represents the atomic ratio of the element.

**Figure 3 polymers-14-04217-f003:**
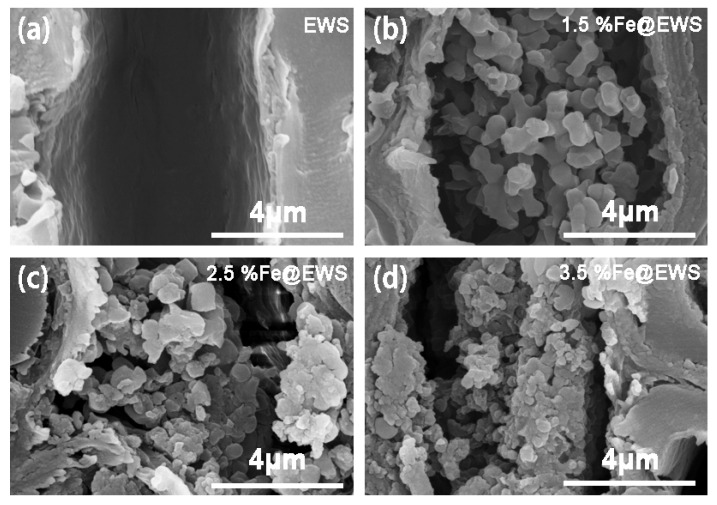
Cross-section SEM images of EWS (**a**), 1.5%Fe@EWS (**b**), 2.5%Fe@EWS (**c**), and 3.5%Fe@EWS (**d**).

**Figure 4 polymers-14-04217-f004:**
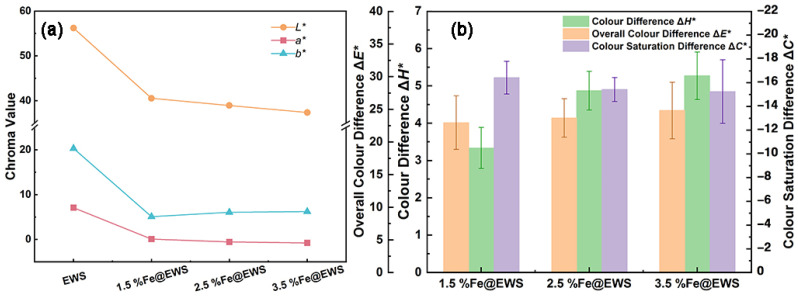
Changes in chromaticity coordinate *a**, *b**, *L** of EWS, 1.5%Fe@EWS, 2.5%Fe@EWS, and 3.5%Fe@EWS (**a**); changes in color saturation difference Δ*C****, color difference Δ*H**, and overall color difference Δ*E** of 1.5%Fe@EWS, 2.5%Fe@EWS, and 3.5%Fe@EWS (**b**).

**Figure 5 polymers-14-04217-f005:**
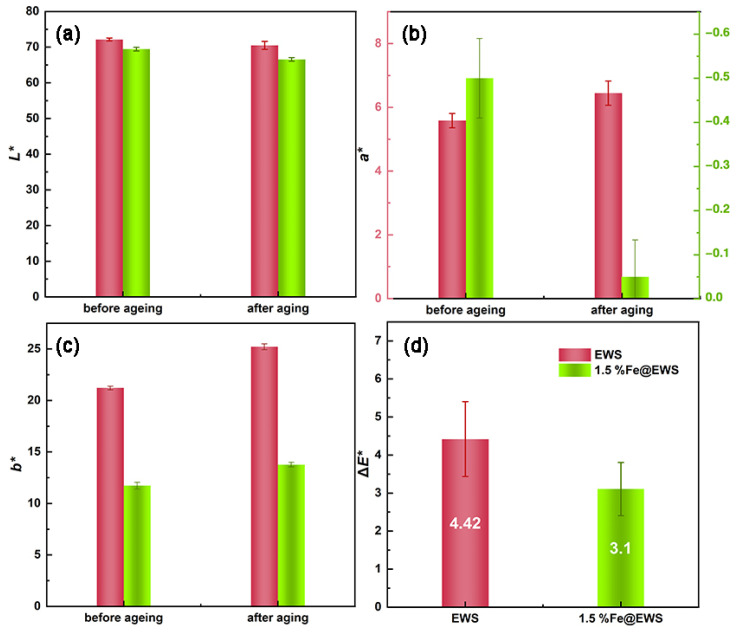
Changes in chromaticity coordinate *L** (**a**), *a** (**b**), *b** (**c**), and Δ*E** (**d**) of EWS and 1.5%Fe@EWS before and after photoaging testing.

**Figure 6 polymers-14-04217-f006:**
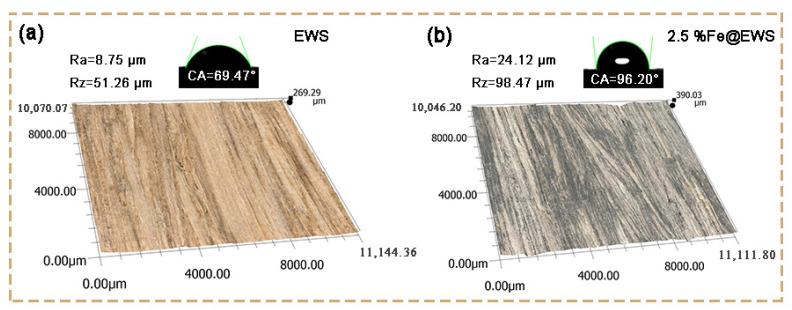
Surface morphologies, roughness, and contact angle of EWS (**a**), 2.5%Fe@EWS (**b**).

**Figure 7 polymers-14-04217-f007:**
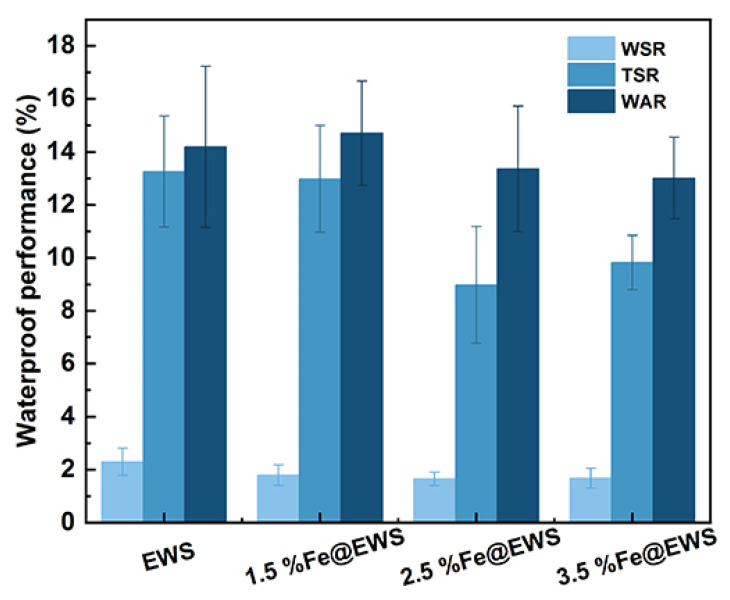
Water absorption rate, thickness swelling rate, and width swelling rate of EWS, 1.5%Fe@EWS, 2.5%Fe@EWS, and 3.5%Fe@EWS.

**Figure 8 polymers-14-04217-f008:**
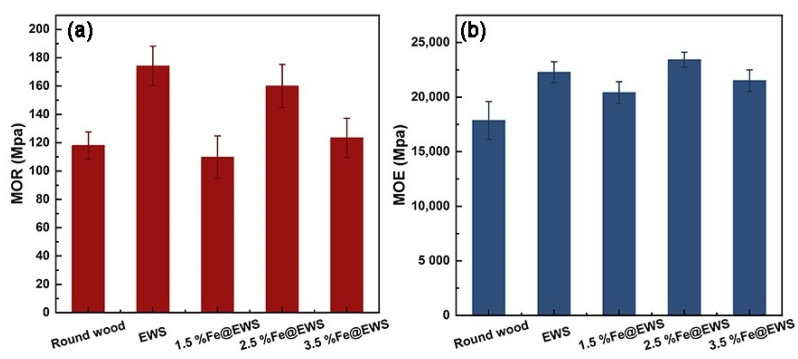
Modules of rupture (MOR) (**a**) and modules of elasticity (MOE) (**b**) of round wood, EWS, 1.5%Fe@EWS, 2.5%Fe@EWS, and 3.5%Fe@EWS.

## Data Availability

Not applicable.

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
