# Peer review of "Preparation and Characterization of Wood Scrimber Based on Eucalyptus Veneers Complexed with Ferrous Ions"

_polymers, 2022, doi:10.3390/polym14194217_

Round 1
Reviewer 2 Report
Comment
In this paper, a new type of wood scrimber based on eucalyptus veneers complexed with ferrous ions was prepared and its properties were characterized. The topic of this paper involves the frontier of the discipline, which has important research significance and a wide range of application prospects. In general, the work is well done, and the conclusion is supported by the experimental and results. However, there are still some issues to be addressed before its acceptance
1. The eucalyptus is a kind of fast growing wood. The author should show the poor physical and mechanical properties of fast growing wood and propose some solutions firstly, and then describe eucalyptus in the second paragraph of Introduction. Some recent references can be cited about the properties of fast growing wood. Such as,
(1) High mechanical properties and excellent anisotropy of dually synergistic network wood fiber gel for human-computer interactive sensors. Cellulose 2022, 29 (8), 4495-4508.
2. The reaction mechanism should be added to Figure 1 so that readers can better understand it. Such as chemical formula.
3. There are two (d) in Figure 1. The author should check them carefully and number them correctly.
4. Figure 3 showed the iron complex in the wood cell lumen. However, there is no iron complex in the lumen. Please explain this. Another question is the iron complex looks big, how does it come into wood cell walls? And whether there is a aggregation occurred of Fe iron during impregnating wood cell walls?
5. The format of Figure 5 should be uniform. (d) data is marked, but (a), (b) and (c) are not marked.
6. The references are old. The author should use more references in recent three years to reflect the present situation of this research field.
7. There are some typos and grammar issues in the manuscript. Authors should carefully recheck the whole manuscript.
Round 2
Reviewer 2 Report
The revised version is good and can be published in this version.